# The Effect of Autogenic Training in a Form of Audio Recording on Sleep Quality and Physiological Stress Reactions of University Athletes—Pilot Study

**DOI:** 10.3390/ijerph192316043

**Published:** 2022-11-30

**Authors:** Kamila Litwic-Kaminska, Martyna Kotyśko, Tadeusz Pracki, Monika Wiłkość-Dębczyńska, Błażej Stankiewicz

**Affiliations:** 1Faculty of Psychology, Kazimierz Wielki University, 85-064 Bydgoszcz, Poland; 2Department of Clinical Psychology, Development and Education, Faculty of Social Sciences, University of Warmia and Mazury in Olsztyn, 10-719 Olsztyn, Poland; 3Institute of Physical Culture, Kazimierz Wielki University, 85-064 Bydgoszcz, Poland

**Keywords:** sleep, sleep quality, autogenic training, athletes, ecological momentary assessment, actigraphy, physiological stress response

## Abstract

Despite the growing popularity of relaxation training, the effectiveness of an autogenic training (AT) as a method of dealing with sleep problems in group of student athletes is unknown. Therefore, this study aimed to fill this gap. University athletes with decreased sleep quality (selected from 209 participants) were randomly assigned to the experimental (EG, *n* = 11) and control (CG, *n* = 11) groups similar in terms of sleep quality, age, gender, type of sport discipline and sport experience. During the 14 days dedicated to performing relaxation training in the form of an audio recording, electronic daily logs and actigraphy were used to monitor the athletes’ sleep and daily activity. The EG listened to the recording with suggestions based on AT and CG only to the background music. Pre- and post-measurements of sleep quality by means of the Pittsburg Sleep Quality Index (PSQI) and physiological stress reactions by biofeedback device were performed. In EG and CG, the parameters of sleep and daily activity obtained by actigraphy and daily logs as well as physiological indicators of emotional reactivity did not differ. Sleep quality in PSQI significantly increased after AT usage in EG. AT seems to be an effective method for university athletes in improving subjective sleep quality, but further studies are necessary.

## 1. Introduction

Sleep plays an important role in physical and mental health [1]. University athletes have to combine both demanding roles of an athlete and a student [2]. Being overloaded with everyday tasks, university athletes may experience worse sleep quality and increased sleep-related difficulties [3]. In university students, these difficulties are generally frequent and affect up to 60% of this population [4]. As for sleep, not only its amount but also its quality is important. According to Buysse, Reynolds, Monk, Berman, and Kupfer [5] sleep quality, as a general construct, should be evaluated in terms of seven aspects related to sleep, which are: “subjective sleep quality, sleep latency, sleep duration, habitual sleep efficiency, sleep disturbances, use of sleeping medications, and daytime dysfunction” (p. 195).

Although physical activity is recognised to improve sleep quality and quantity [6,7], the prevalence of poor sleep quality—as measured by the Pittsburgh Sleep Quality Index (PSQI)—is also quite high among athletes, reaching up to 64% [8,9,10,11]. It should be noted that the definition of poor sleepers varies depending on the PSQI score criterion adopted: ≥5 or >5 [8]. Difficulties with sleep may apply to athletes due to a number of sport-related requirements, such as acute (e.g., playing away, excitement) and chronic stressors (e.g., different match schedules) resulting from intensive trainings and competitions [12,13]. Based on research to date, it can therefore be assumed that the prevalence of sleep problems and reduced sleep quality among athletes is significant and cannot be ignored. This is also pointed out by athletes themselves and their coaches, while indicating that sleep-related difficulties may cause short-term fatigue [14]. Since the athletes may be prone to sleep problems it is worthwhile to verify the efficacy of different types of sleep interventions [8,15,16].

As an alternative to pharmacological interventions, usage of relaxation techniques turns out to be an effective method of improving the sleep quality in the case of functional sleep disturbances [17]. Literature review indicated that relaxation techniques helped to improve the quality of sleep (less awakenings at night, longer sleep time) and decrease the sense of fatigue after waking up in oncological patients, where sleep problems arise as a consequence of the disease e.g., [18], in older adults [19] and were shown to be an effective, non-pharmacological treatment for insomnia [20]. Relaxation methods provided medium effects for sleep quality and sleep problems in college students [21]. There are many reports showing that athletes used the relaxation techniques to reduce competitive anxiety or to improve athletic performance, e.g., [22]. However, there is a lack of studies on the use of relaxation techniques by athletes aimed at improving the quality of sleep. To date, we have found only one article presenting an experimental study on a very small sample (*n* = 12) of female football players [23].

Two broad reviews [21,24] showed that among the relaxation techniques applied in research on improving the sleep quality were, among others, progressive muscle relaxation, mindfulness meditation, biofeedback, relaxing music, or autogenic training (AT). When searching for an effective method to improve sleep quality in a group of student-athletes, we chose Schulz autogenic training. Review of studies in groups of students [21] and meta-analysis of clinical outcome studies [17] showed that AT is an effective method to help counteract sleep problems by improving the quality of sleep, sleep latency, the duration of sleep and the energy level after waking up. We assumed that the relaxation training enhances subjective sleep quality and sleep timing parameters also in university athletes. Taken into account the results of our previous study on the interrelation between sleep quality and stress [25] and reports indicating the impact of AT on stress response [26,27] we also controlled the physiological response to stress.

Thus, the main aim of this research was to evaluate the effect of the AT on sleep quality and physiological stress reactions in a group of university athletes.

## 2. Materials and Methods

### 2.1. Participants

Two hundred and nine students from one of the Polish universities actively involved in sport (i.e., active membership in a sport club or academic sport association, where participants train or have an individual training routine) took part in the recruitment process and were screened for eligibility. Exclusion criteria was a history of chronic disease that might affect circadian rhythms, or cause fatigue, for example anaemia, asthma, diabetes, depression and related to it medication intake. Gupta et al. [7] in the research review on sleep quality among elite athletes adopted three cut-off values based on PSQI scores, i.e., ≥5, >5 and >8. In our study, the inclusion criteria were set at PSQI score ≥ 5. After rejecting participants due to incomplete questionnaires (*n* = 2), 85 athletes with reduced sleep quality were invited to participate in the experimental part of study. Thirty-one participants who accepted the invitation were allocated to the experimental (EG; including 6 women and 10 men) or control group (CG; including 6 women and 9 men). The data collected from 9 participants were excluded because an insufficient number (less than 6) of relaxation sessions were conducted (*n* = 7) or the participant did not report for the post-measurement (*n* = 2). As a result, the full data were obtained and analysed from 22 athletes—equally for EG and CG. The selection process is shown on a flowchart in Figure 1.

### 2.2. Procedure

The presented study was an experimental research of randomized block design with single blind repeated measures. Pre- and post-measurements were conducted two weeks apart to check for changes in sleep quality (using the PSQI) and emotional reactivity (using biofeedback devices). Physiological parameters were measured in a sitting position and at the same time of day so that the circadian rhythm did not affect physiological data. All the measurements were conducted by principal investigator and two trained final year psychology students, in laboratory—severe room, where only table with biofeedback device, relaxing chair and metal wardrobe was installed.

During the two-week period, the Ecological Momentary Assessment (EMA) procedure was used. Each participant wore an actigraph, completed the daily log (by Android application) and listened to the recording (implemented in the application). Experimental group (EG) had to listen to the recording with the relaxation training (based on the AT method), control group (CG) listened to the background music without suggestions, containing only introductory and final instructions (e.g., which position of the body to take). Both groups were supposed to perform themselves the training every day in the evening and not right after physical activity during the two-week period of intervention.

The study was conducted according to the guidelines of the Declaration of Helsinki and was approved by the university ethics committee for scientific research. Before the study, all the students were given a written study information and filled the informed consent.

### 2.3. Instruments

#### 2.3.1. Sleep Quality

The PSQI [5] as a retrospective method assesses sleep quality over a one-month time interval. It includes 19 items that allow to distinguish seven components which are summed to produce a global score. Results above 5 points indicate “poor sleep” [5], however in the study cut-off criteria were set at five or more points according to Samuels study [28].

#### 2.3.2. Physiological Parameters

Physiological stress reactions were measured using common non-invasive techniques for detecting stress [29]. Measurements were taken with the Biofeedback device: Biograph Infiniti, Thought Technology. The Blood Volume Pulse with Heart Rate (BVP+HR), Skin Conductance (SC), Temperature was recorded using four sensors attached to the fingers. Additionally, a breath measurement was included. Respiration sensor was placed around the abdomen. Each of the participants underwent the same procedure according to the task force’s recommendation [30]. The experimental design (both pre- and post-measurements) containing three 5-minutes’ stages: baseline (anticipation of stress), event (arithmetic task-induced stress) and post-event (post-activity rest).

#### 2.3.3. Actigraphy

Actigraphy is indicated as a reliable tool to study the impact of treatments to improve sleep [31]. The Actiwatch AW4 (Cambridge Neurotechnology, Cambridge, UK) and the Actiwatch 2 (Philips Respironics, Hong Kong, China) actigraph wrist watches were used to monitor continuously movement activity (excluding the time of training, bathing, and the circumstances in which physical damage to the equipment could occur) for two weeks. Before the research all watches were calibrated, and the subjects used the same devices in subsequent measurements. A 2 min epoch length was applied. The actigraphy analysis included sleep duration.

#### 2.3.4. Daily Logs—ADS Application

A self-constructed Android application was used to collect data about athletes’ daily activity that may influence sleep quality and quantity (undertaken physical activity and its intensity—on a 5-point Likert scale) and to check the sleep parameters (sleep duration, and in a Likert scale survey: subjective sleep quality, level of energy, level of stress).

#### 2.3.5. Relaxation Training

Self-created audio recording SAT-relax that is based on the conception of the Schultz’s AT was used [32]. It involves suggestions related to passive concentration of bodily perceptions of heaviness and warmth as well as a slow breath which are supposed to induce a state of relaxation [33]. Nine-minute recording was implemented in the ADS application. Background music used in the recording was selected based on the assessments of competent judges and purchased from https://audiojungle.net/item/nature-ambient/14431334?s_rank=6. (accessed on 8 February 2017)

### 2.4. Data Analysis

Data were analysed with IBM SPSS v28 (IBM, Armonk, NY, USA). Initial comparison of EG and CG included: sex ratio and sport discipline (chi square test with Yates’ correction), age and number of relaxation sessions (*t*-test for independent samples). Due to non-fulfilment of the assumption of normality of distribution for multiple variables, non-parametric statistics were used. Main comparisons between the groups were analysed with Mann–Whitney *U* test. Data collected by actigraphy and daily logs were organized into pre- (first four nights) and post- (last four night) measurements. All pre- and post-measurements were analysed with Wilcoxon signed-rank test (for dependent samples). Correlation coefficient *r* was used as an effect size (ES) for Wilcoxon test. It was calculated with the formula: $r=Z/\sqrt{n}$ [34] and interpreted according to Cohen’s guidelines [35]. Spearman correlation coefficient was used to verify relation between the number of relaxation sessions, time spent on physical activity and its mean intensity, and change in subjective sleep quality from PSQI (delta calculated as the difference between pre- and post- measurement).

## 3. Results

The result of the chi square test confirmed that the groups did not differ in terms of sex ratio (EG: Women, *n* = 3, Men, *n* = 8; CG: Women, *n*= 6, Men, *n* = 5; *Χ*^2^ = 0.752, *p* = 0.386). Additionally, age (M_EG_ = 22.36, M_CG_ = 21.73, *t* = 0.725, *p* = 0.477) and number of relaxation sessions (M_EG_ = 11.00, M_CG_ = 10.36, *t* = 0.530, *p* = 0.602) was similar in compared groups. In both EG and CG, the same number of participants were representatives of individual (*n* = 7) and team sports (*n* = 4).

The main comparison between EG and CG includedsleep quality measured with PSQI, sleep parameters collected via actigraphy and daily log, and physiological data. Detailed information is shown in Appendix A. According to the analysis, both groups, when independently compared with Mann–Whitney *U* test, presented similar results in all analysed pre and post measurements. Differences were noted in EG pre-post dependent results, where PSQI score significantly decreased—sleep quality improvement (*Z* = 2.54, *p* = 0.011, ES: *r* = 0.54) and sleep duration obtained via daily log increased (*Z* = 2.13, *p* = 0.033, ES: *r* = 0.45). The presented values of ES were large and medium, according to Cohen’s interpretation suggestions [35]. Change in pre-post PSQI subjective sleep quality (delta) in both groups was correlated with the number of relaxation sessions. A statistical tendency was present in EG, where *R* = 0.53, *p* = 0.093, but not in CG, *R* = 0.12, *p* = 0.728. Mean time spent on physical activity and its intensity during the 2-week period were correlated also with PSQI delta. Among EG no statistically significant relations were noted, but in CG the relationship between change in sleep quality and physical activity intensity was significant (*R* = −0.63, *p* = 0.038).

## 4. Discussion

In our study, we checked the effects of AT on sleep quality and physiological stress responses of university athletes in an experimental procedure. The results showed that in the EG (where AT was used) the overall sleep quality measured with PSQI increased. In the CG (where only background music was used), there was no significant change between pre- and post-measurement. We can assume that the AT, as a relaxation method can be more effective in enhancing the subjective assessment of the sleep quality than the background music alone. It is consistent with the basic assumptions of AT that it may be used to relax the body by relaxing the mind [33].

According to the meta-analysis carried out by Stetter and Kupper [17] in clinical groups, AT should be treated as an add-on support to medical treatment. The outcome presented in this article shows that treatment with AT can be sufficient for healthy people, such as academic athletes for enhancing their subjective quality of sleep. The effect of AT on sleep quality was large which shows that AT, as an alternative to pharmacological interventions (medications or supplements), might play an important role in helping academic athletes improve their sleep quality. This is a positive conclusion, considering that such relaxation trainings are an accessible form of impact and (especially in the shape of audio recording) are easy to use and seem to have no side effects. It should also be mentioned that changes in sleep quality after AT in the EG occurred independently of the amount and intensity of physical activity. In contrast, in the CG, the greater intensity of physical activity was connected with the reduced change in PSQI score.

Concerning the lack of differences in most of the objective indices of sleep as well as physiological stress responses, we assume a few possible explanations of that. Firstly, the problems with sleep examined in the experiment group were rather mild (8 points was the highest noted result in PSQI).

Secondly, the duration of intervention was quite short. However, our results are in line with Schlarb, Friedrich and Classen [36], where also a 2-week training (CBT and hypnotherapy) was implemented among university students, and similarly the change was recorded in the PSQI but not in actigraphy. Generally, there are large discrepancies in the duration and frequency of relaxation trainings in prior studies [17,21,24]—from very brief of a few hours, through intense, everyday sessions for a couple weeks (as in our study) to several months of two or three times a week training sessions. We are aware that the effectiveness of the intervention may increase with its duration [37]. Relaxation training is a long-term process requiring regularity, as it aims to fully automate the reactions [33]. Perhaps a longer period of training application would allow to collect more data on its effectiveness. The presented research was a pilot study, so we used the shorter period reported in the literature. Therefore, it may provide indications for the design of subsequent interventions. To some extent, this has a reference in the trend noted in our study, where, in the experimental group, the change in PSQI (sleep improvement) was positively related (at the level of statistical tendency) to the number of training sessions completed.

Thirdly, the lack of significant change in physiological parameters may be due to the fact that the implementation of AT is not aimed at changing specific physiological parameters (as in biofeedback trainings [38]), but rather at subjective feelings of calmness and relaxation.

The results may inspire athletes and people working with them to include the AT in mental trainings, not only in case of dealing with competitive anxiety or improve athletic performance (the efficiency of which has been previously proven), but also in case of some problems with sleep.

The presented study is not free from limitations. The major is the lack of the control group without any intervention (any relaxation training between pre- and post-measurements). Therefore, we can demonstrate only the effect of suggestions based on autogenic training and we cannot conclude about the effectiveness of the music itself. Due to the non-normality of the distribution, it was not possible to perform an ANOVA which would have given a more complete possibility of inference.

The subjects did not wear actiwatches all the time. We controlled, by the daily logs, the reasons for taking off the watches. Moreover, the students of subsequent years had different class schedules covering different number of physical activities. According to these circumstances, it was checked whether there were no differences between EG and CG in terms of the initial level of sleep quality, length of sleep, time spent on physical activity and its intensity. Lack of significant differences between groups strengthens the obtained result in the use of relaxation training as a form of intervention for sleep quality improvement.

## 5. Conclusions

The presented preliminary results found that autogenic training may be used as a tool that improves the sleep quality in healthy university athletes more than the relaxing background music alone. However, further analyses, in a larger group, considering such variables as sports level or type of sport discipline, are necessary.

## Figures and Tables

**Figure 1 ijerph-19-16043-f001:**
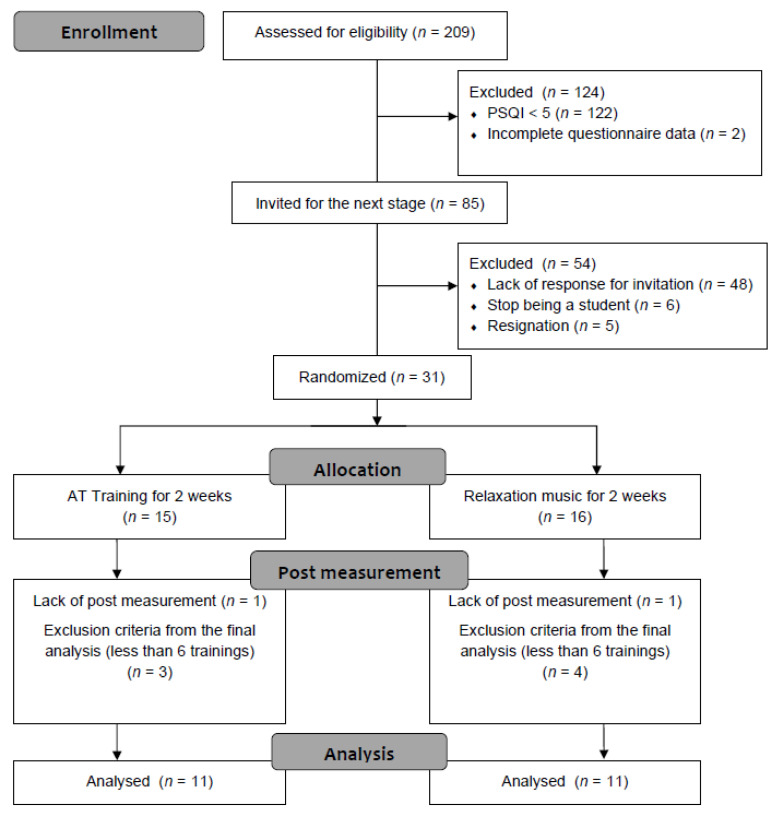
Participants flowchart.

## Data Availability

The data presented in this study are available on request from the corresponding author.

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
