# Peer review of "The Effect of Autogenic Training in a Form of Audio Recording on Sleep Quality and Physiological Stress Reactions of University Athletes—Pilot Study"

_ijerph, 2022, doi:10.3390/ijerph192316043_

Round 1

Reviewer 1 Report

It is a relevant topic - testing a potentially effective intervention to improve sleep quality in a population that often suffers from sleep problems. However, I believe there are several important limitations in the study design as well as several reporting problems throughout the manuscript.

Introduction:

Line 33: what do you mean by “healthy factor”?

Line 40: I believe it should be “the definition of poor sleep quality” rather than “the data about poor sleepers”.

Line 44: do you consider napping a “sport-related requirement”?

Line 45: unclear and vague sentence.

Line 49: do you mean “quality” or “efficacy”?

Line 50: “what was raised by scientist” – is vague and I believe it can be removed from the sentence.

Line 54: note that total sleep time and sleep quality are separate constructs.

Line 62: did you search only on the Web of Science database?

Line 63: “experimental study” or “experimental design”?

Line 68: “chosen” by whom? By you or by the studies you mentioned?

Line 70: “solving sleep problems” is a very strong statement.

Your objectives do not reflect what you have done. It seems that you are not looking at the effect of one intervention, but comparing two interventions as you have stated in your title and discussion.

The rationale structure is good; however, there are several grammatical and subject-verb agreement errors throughout the section that should be revised.

Materials and methods:

Line 88: why did you choose this cut-off value?

Lines 92-95: you should describe whether there were differences in baseline characteristics between those who were lost and those who were included in the analysis. Differences in sleep parameters, for instance, could indicate an important source of bias and may be related to the follow-up loss/lower adherence to interventions. Moreover, an intention-to-treat approach would be desirable rather than just excluding those who did not adhere to interventions. Also, what did you consider as an “insufficient number of sessions”?

102: were the assessors blinded? It is unclear.

Flowchart: it is saying that you excluded 122 participants for presenting a PSQI score ≥5, which was actually one of the inclusion criteria.

Line 123: it is important to mention that PSQI is a retrospective questionnaire.

Line 132: “a breathing was attached” – more clarity is needed.

Lines 139-142: for how many days was the actigraph used? It is important information and should be stated in your methods. Moreover, what variables did you extract from the actigraphy data?

Line 145: is there any reference to support the use of this application?

Line 151: how many sessions were done and what was the frequency of intervention delivery? Who delivered the interventions?

I missed information on what the CG did.

Please revise some grammatical and subject-verb agreement errors in this section.

Results:

Line 175: it makes no sense to perform statistical analysis to test for between-groups differences in baseline characteristics. The role of the randomization process is to ensure similarity between groups. The p-value does not add much information in this case, for instance, you did not find statistical differences, however, the number of women was double in the CG group compared to the EG. The SD would be more informative than the p-value when reporting mean values. Moreover, it would be informative to have baseline data on physical activity level, training frequency, etc.

Line 185: I found your results very conflicting. The PSQI values in pre- and post-interventions were the same in both groups (6 and 5, respectively), therefore, how the EG post-pre difference was statistically significant (p=0.01) and not significant in the CG (p=0.10)?

Line 188: can you provide the values from the correlation analysis?

Line 190: what did you mean by "slight tendency"? the result was not statistically significant (p>0.05).

Discussion:

Line 196: you cannot make this assumption based on post-pre analysis alone.

Line 198: what did you mean by “effectiveness of the suggestions included in the recording”? What suggestions?

Line 199: I do not believe that this assumption can be made.

Line 205: what was your definition of “large effect”?

Line 206: it is a very strong statement to make based on your study design and your findings.

Lines 210-211: a very short paragraph like this should be avoided.

Lines 212-216: I did not understand how the fact that most participants were excluded for having good sleep quality could be a fair explanation for the lack of differences found in objective sleep parameters and stress evaluation.

Line 222: if the previous literature suggests that it should be a long-term intervention, why did you choose this 2-week duration?

Lines 228-232: more clarity is needed.

Line 239: again, it seems different from your aims. Investigating the effect of an intervention is different from comparing the effects of two or more interventions.

Another important limitation and source of bias in your study is the large difference between groups in baseline sleep duration recorded by actigraphy (nearly 26 min of difference).

Reviewer 2 Report

The paper aim to check the effects of autogenic training versus music relaxation on sleep quality and physiological stress responses of university athletes in a two-week experimental procedure. The topic is interesting and the paper is well written. 

Line 117-118. If you have a reference number from the ethics committee for scientific research, please add it.

Line 157. Authors should consider if the link of a music that is possible we cannot listen to is necessary. Only the website and the sound category may be necessary.

Data analysis and results. How do the data related to physical activity, the intensity of physical activity and the stimulants or energy drinks ingested (line 148) influence or not in the results?

The manuscript presents a substantial contribution to the literature on the use of autogenic training to improve the sleep quality. I encourage the authors to continue this research, increasing the training period and increasing the study sample.

Round 2

Reviewer 1 Report

The revision significantly improved the clarity of the manuscript.